# Antifungal Activity of Phyllospheric Bacteria Isolated from *Coffea arabica* against *Hemileia vastatrix*

**DOI:** 10.3390/microorganisms12030582

**Published:** 2024-03-14

**Authors:** Katty Ogata-Gutiérrez, Carolina Chumpitaz-Segovia, Jesus Lirio-Paredes, Doris Zúñiga-Dávila

**Affiliations:** Laboratorio de Ecología Microbiana y Biotecnología, Departmento de Biología, Facultad de Ciencias, Universidad Nacional Agraria La Molina, Av. La Molina s/n, Lima 15024, Peru; kogata@lamolina.edu.pe (K.O.-G.); jlirio1@icloud.com (J.L.-P.)

**Keywords:** bacteria, phyllosphere, biocontroller, coffee rust, biotroph

## Abstract

Peru is one of the leading countries that produce and export specialty coffees, favorably positioned in the international markets for its physical and organoleptic cup qualities. In recent years, yellow coffee rust caused by the phytopathogenic fungus *Hemileia vastatrix* stands out as one of the main phytosanitary diseases that affect coffee culture yields. Many studies have demonstrated bacteria antagonistic activity against a number of phytopathogen fungi. In this context, the aim of this work was to select and characterize phyllospheric bacteria isolated from *Coffea arabica* with antagonistic features against coffee rust to obtain biocontrollers. For that purpose, a total of 82 phyllospheric bacteria were isolated from two coffee leaf rust-susceptible varieties, *typica* and *caturra roja*, and one tolerant variety, *catimor*. Of all the isolates, 15% were endophytic and 85% were epiphytes. Among all the isolates, 14 were capable of inhibiting the mycelial radial growth of *Mycena citricolor*, and *Colletotrichum* sp. 16S rRNA gene sequence-based analysis showed that 9 isolates were related to *Achromobacter insuavis*, 2 were related to *Luteibacter anthropi* and 1 was related to *Rodococcus ceridiohylli*, *Achromobacter marplatensis* and *Pseudomonas parafulva*. A total of 7 representative bacteria of each group were selected based on their antagonistic activity and tested in germination inhibition assays of coffee rust uredinospores. The CRRFLT7 and TRFLT8 isolates showed a high inhibition percentage of urediniospores germination (81% and 82%, respectively), similar to that obtained with the chemical control (91%). An experimental field assay showed a good performance of both strains against rust damage too, making them a promising alternative for coffee leaf rust biocontrol.

## 1. Introduction

Coffee is one of the most important crops worldwide and is considered to be one of the main commodities for many developing countries [1]. In Peru, coffee is one of the main agricultural products exported and represents 86% of Peru’s total traditional exports. Coffee registered a 3.3% increase in its exportation value compared to that in 2019, due to an improvement in the average price during the last 4 months of the year [2]. Furthermore, Peru, along with Ethiopia, is currently the leading global exporter of organic coffee beans in the world [3] after Mexico and is the main exporter to The United States. Peruvian specialty coffees are recognized for having one of the best physical cup qualities and are becoming famous in many countries, mainly in The United States, Germany and Belgium [4]. Peru has about 425,000 ha of cultivated coffee, representing 10.2% of the national agricultural area with crops. There are currently 236,000 producers involved in the production of coffee, which means that close to 1 million families depend on its cultivation, and there are more than 660,000 ha of coffee cultivated [5]. However, many plant diseases could limit coffee production, causing high economic losses. Coffee leaf rust is one of the most important and aggressive diseases that devastate coffee plants in short periods. Between 2012 and 2013, this biotrophic fungus, *Hemileia vastatrix*, affected about 400,000 ha of coffee cultures, which means a loss of about 414 million soles [6], causing a great impact on coffee farmers whose economy depends mainly on its exportation. The control of this plague could be handled with chemical fungicides and with an adequate management of plant fertilization. However, the indiscriminate use of chemicals and fungicides in crops is an increasing problem since international markets have strict regulations concerning the presence of these chemicals’ residue in food and commodities of plant origins. Agriculture is one of the main sources of water and soil pollution [7], causing negative environmental and health [8] effects too. In this context, agriculture based in a sustainable and friendly environment becomes necessary. Microbial inoculants are a natural and organic alternative to chemicals and could be used as biofertilizers and biocontrol agents. Biological control is considered a potent tool for reducing damage caused by certain pathogens; although its performance is not expected to be as effective as chemical fungicides, numerous authors report its efficacy against some phytopathogens. In this sense, it can provide an interesting alternative to fight against coffee diseases sustainably. Bacteria are reported to be the most abundant group of microorganisms that colonize leaves, with culturable counts, growing in a range between 10^6^ and 10^11^ per gram of leaf [9]. Because rust is a pathogen that primarily infects coffee leaves, causing plant defoliation, determining the microorganisms that make up the phyllosphere becomes important in plant biocontrol. The phyllosphere is defined as the total aerial area of a plant, mainly leaves; this can serve as a habitat for different microorganisms [10]. Phyllosphere bacteria in this context are the most abundant and may have the potential to suppress some important coffee diseases, including that caused by the phytopathogenic fungi *H. vastatrix*. Bacteria have different mechanisms to lead plant biocontrol, including the production of antagonistic metabolites or certain degradative enzymes, competition and the induction of host resistance, among others. Based on this information, this work pretends to select and characterize coffea phyllospheric bacteria with antagonistic ability against *H. vastatrix* to obtain potential biocontrollers of this biotrophic fungi.

## 2. Materials and Methods

### 2.1. Isolation of Bacteria

Bacteria were isolated from leaf samples of *Coffea arabica* of three varieties: two susceptible (*typica* and *caturra roja*) ones that lack genetic resistance to yellow rust, and one tolerant to rust (*catimor*) because it is a hybrid of *timor* (*C. canephora*) and *caturra* (*C. arabica*) varieties. *typica* and *caturra roja* Samples were collected from the Instituto Regional de Desarrollo of the jungle (IRD-Selva), Genova-UNALM, which is located in Chanchamayo-Junin (11°05′42.6298″ S, 75°21′9.4349″ W). Ten leaves of five randomly selected coffee plants were selected for each variety, packed in hermetically sealed bags and delivered into a cooler to the laboratory Laboratorio de Ecología Microbiana y Biotecnología (LEMyB)—UNALM in Lima, Peru for the forward analysis.

### 2.2. Determination of Phyllospheric Bacterial Populations

The quantification of the endophytic bacteria was carried out according to the methodology proposed by [11]. Leaves were previously disinfected with 70% ethanol and washed with 1% sodium hypochlorite, both for 2 min. Then, they were rinsed with sterile distilled water and macerated in a mortar with a pestle, using 1X PBS buffer in a proportion of 1:20 (*w*/*v*). Serial dilutions of the maceration were made using 0.85% NaCl. Yeast extract mannitol medium (YEM) agar (mannitol 10 g/L, yeast extract 0.5 g/L, K_2_HPO_4_ 0.5 g/L, MgSO_4_·7H_2_O 0.1 g/L, NaCl 0.2 g/L, agar 15 g/L) was used for bacterial growth, quantification, and isolation. Plates were incubated for 15 days at 28 °C after being processed. Meanwhile, the quantification of total epiphytic aerobic bacteria was made using the whole sampled leaves and by washing them with sterile distilled water to clean the excess dust. Subsequently, each leaf was rinsed in a sterile flask with 1X PBS buffer in a proportion of 1:20 (*w*/*v*) and was shaken for 3 min. The spread plate technique in YEM agar was used to obtain epiphytic bacteria for each leaf suspension and its dilutions. The plates were incubated at 28 °C for 15 days to be evaluated. Additionally, both of the PBS suspensions obtained from the endophyte’s and the epiphyte’s methodologies were treated to obtain spore-forming bacteria too, using the methodology proposed by [12]. For this purpose, dilutions were treated in a water bath at 80 °C for 20 min. Subsequently, 1 mL of each dilution was served in Petri dishes with Tryptone glucose extract (TGE) medium (triptone 5 g/L, meat extract 3 g/L, D-glucose 1 g/L, agar 15 g/L), using the spread plate technique. The plates were incubated at 28 °C for 24 h. The quantification of the bacteria population was reported as colony forming units (CFU)/mL of bacterial culture media. Colonies that showed morphological differences from each other in all the media used were selected and isolated.

### 2.3. In Vitro Antifungal Activity Assay against M. citricolor and Colletotrichum sp.

Antagonistic in vitro assays of the strains against *M. citricolor* and *Colletotrichum* sp. were performed using the dual plate method [13]. A mycelial disk of each fresh fungi culture was placed in the central area of the Potato Dextrose Agar (PDA) (potato extract 4 g/L, Dextrose 20 g, agar 15 g/L) plate and grown at 28 °C for 5 days. In total, 3 µL of each bacterial culture (10^8^ CFU/mL) grown in Nutrient Broth was placed at an equal distance of 3 cm from the center of the Petri dish where the fungus was growing. The experiment was Completely Randomized Design (CRD) and carried out in triplicate. Strains that inhibit mycelial growth were considered positive. The plates were incubated in the dark at 28 °C and evaluated daily thereafter for 10 days post-inoculation. The percentage of mycelial growth inhibition was calculated using the following formula: FGI (%) = (R − r)/R) × 100, where R represents the radius of the fungus without any treatment, and r represents the radius of the fungus growing with the evaluated strain (Idris et al., 2007 [13]).

### 2.4. DNA Extraction and 16s Ribosomal RNA Gene-Based Phylogenetic Analysis

Fourteen isolates were selected based on their antifungal activity against *M. citricolor* and *Colletotrichum* sp. The genomic DNA of bacterial cultures was extracted using the AxyPrep Bacterial Genomic DNA Miniprep Kit (Axygen Scientific, Union City, CA, USA), according to the manufacturer’s instructions. 16S ribosomal RNA gene amplification (1500 bp) was performed using the primers fD1 and rD1 [14].

The PCR reaction mix (25 µL) contained: 1X reaction buffer (Fermentas, Waltham, MA, USA), 2 µL (50 ng) of extracted genomic DNA, 1.5 mM MgCl_2_, 5 pmol of each oligonucleotide, 200 µM dNTPs (Fermentas, Waltham, MA, USA) and Taq polymerase (1U/µL, Fermentas, Waltham, MA, USA). The PCR temperature cycling conditions were as follows: initial denaturation at 94 °C for 3 min, followed by 35 cycles of denaturation at 94 °C for 45 s, annealing at 57 °C for 1 min and elongation at 72 °C for 2 min. The last cycle was followed by a final extension at 72 °C for 5 min. The amplification products were examined using a 1% agarose gel and purified with the AxyPrep TMR PCR Cleanup Kit (Axygen Scientific, Union City, CA, USA) according to the manufacturer’s instructions and subsequently sequenced by a commercial service (Macrogen Inc., Seoul, Republic of Korea). The obtained sequences were examined and edited using the BioEdit sequence alignment program [15] and identified through BLASTn (Basic Local Alignment Tool for nucleotides) from the public database of The National Center for Biotechnology Information (NCBI) (http://www.ncbi.nlm.nih.gov (accessed on 28 November 2023)). Multiple alignments were compared using the Clustal X2 software [16]. Phylogenetic analyses were performed using the neighbor joining (NJ) method [17], with Mega6 software, applying 1000 bootstrap test subsets with genetic distances computed using the Tamura Nei model [18]. This analysis was performed with the isolates that demonstrate an inhibitory effect against the phytopathogenic fungi.

### 2.5. Inhibitory Effects of Phyllospheric Bacterial Isolates on the In Vitro Germination of H. vastatrix Urediniospores

The germination inhibition capacity of selected bacteria against coffee rust urediniospores in vitro [19,20] was tested. Bacterial isolates from the phyllosphere were cultivated in SGM growth medium (Na_2_HPO_4_ 0.5 g/L, Na_2_MoO_4_ 0.6 g/L, KH_2_PO_4_ 0.3 g/L, NaCl 0.1 g/L, MgSO_4_·7H_2_O 0.2 g/L, CaCO_3_ 0.022 g/L, iron citrate 3.8 mM 5 mL/L, microelements 1 mL/L, yeast extract 0.5 g/L and sugar 5 g/L, pH 7.3 ± 0.1) for 72 h at 150 rpm and 28 °C. From one side, each bacterial concentration was adjusted with a 0.85% saline solution until 10^8^ CFU/mL [21,22]. On the other side, coffee rust urediniospores obtained from plants showing disease symptoms were collected with a sterile scalpel. The urediniospores were resuspended in sterile distilled water supplemented with 0.001% tween 80, and the concentration was adjusted to 10^5^ urediniospores/mL. In total, 10 µL of bacterial culture and 10 µL of the urediniospores suspension were placed on a concave slide inside a humidity chamber system and incubated in the dark at 23 ± 0.2 °C for 48 h. The commercial fungicide propiconazole was used as a positive control against the *H. vastatrix*. For this assay, the propiconazole concentration was 0.3 g/L, according to the manufacturer’s recommendation. The experiment was repeated twice, and each treatment consisted of three replicates.

### 2.6. Foliar Application of Selected Isolates in Coffee Plants var. Caturra Roja

A field experiment was carried out in June at IRD (Instituto de Desarrollo Regional)-jungle Genova, near Santa Rosa annex (750 m altitude, 11°5′40.634″ S, 75°21′ 50.424″ W). Around 40 six-month-old, inoculated coffee seedlings var. *caturra roja* were transplanted from the nursery to the field. The experiment was configured in a 1.5 m row-spacing configuration, and the plants were planted at 1 × 1 m spacing. To avoid the edge effect, additional coffee plants were transplanted around the perimeter. Selected bacteria inocula (CRRFLT7 and TRFLT8) were tested independently. Propiconazole was used as the chemical control and water was used as the negative control. The bacteria inocula of CRRFLT7 and TRFLT8 isolates were prepared using the SGM broth with a concentration of 10^8^ CFU/mL. Each treatment was applied by spraying them on the foliar and the abaxial zone every fifteen days (Appendix A). The damage level of the disease on the different treatments was recorded twice, 7 and 8 months after the transplant, when symptoms appeared. Natural infection with *H. vastatrix* in each treatment was assessed to determine plant damage. The disease incidence percentage (DI%) [23] and disease severity expressed as the pathogen disease index (PDI) were the parameters evaluated, calculated as the following formula:(1)$$\mathrm{DI}\%=\frac{Number\;of\;infected\;leaves}{Total\;number\;of\;leaves}\times 100$$
(2)$$\mathrm{PDI}=\frac{Sum\;of\;all\;numerical\;grade}{Total\;number\;of\;leaves\;counted\times Maximum\;grade}\times 100$$

PDI was calculated using the 0–5 scale of [22] with the following scores: 0 = no symptom; 1 = 0.1–5.0% leaf area affected (l.a.a.); 2 = 5.1–15.0% l.a.a.; 3 = 15.1–30.0% l.a.a.; 4 = 30.1–50% l.a.a.; 5 = 50.1–100% l.a.a. All data were analyzed by an LSD ANOVA multiple range test with ten replicates per treatment.

## 3. Results

### 3.1. Microbial Population and Isolation of Phyllospheric Bacteria

The population of endophytic and epiphytic bacteria (Table 1) was quantified as the number of colony-forming units (CFU) per gram of dry weight leaf samples obtained from the coffee plants var. *catimor*, *caturra roja* and *typica*. The abundance of the populations varies according to the group of bacteria studied (Table 1). The varieties *catimor* and *typica* showed endophytic bacteria in the leaf samples, compared to the *caturra roja* variety, in which no endophyte was recovered. In addition, leaves of the symptomatic *typica* variety showed the highest population of endophytes (21 × 10^5^ CFU·g^−1^ leaf). On the other hand, the aerobic epiphytic bacteria population varies between 10^3^ and 10^9^ CFU·g^−1^ leaf, the symptomatic samples of *catimor* and *typica* being those that showed a higher population compared to the asymptomatic ones (Table 1). It was observed that symptomatic samples of the *typica* variety showed the highest population in both endophytic and epiphytic aerobic bacteria. Furthermore, the presence of sporogenic aerobic epiphytic bacteria was different in each variety, showing a higher population (23 × 10^2^ CFU/mL) in asymptomatic samples of the *catimor* variety compared to the other samples (Table 1). Based on the cultivable bioprospecting, a total of 82 bacterial strains were isolated from three coffee plant varieties. From these isolates, 11 were endophytes (13%) and 71 were epiphytes (87%). Within this last group, 51% and 49% belonged to the aerobic and sporogenic bacteria, respectively. The bacterial population size depends on the leaf zone, plant variety, leaf health and group of bacteria studied.

### 3.2. Selection of Fungal Antagonists Using Coffee Phytopathogenic Fungi

According to the results, the CSEDT7 isolate was the only endophyte that exhibited an in vitro antagonistic activity against *M. citricolor* and *Colletotrichum* sp., showing 42% and 20% inhibition of radial growth for each fungi, respectively (Table 2). In the same way, thirteen aerobic epiphytic isolates were capable of significantly inhibiting the development of both fungi. It was observed that all the isolates showed the highest antagonistic activity against *M. citricolor* with an inhibition percentage of more than 33%, CSFLT4 and CSFLT5 showing the best results (44%). When the isolates were confronted against *Colletotrichum* sp. instead, their inhibition capacity dropped. It was observed that the endophytic bacterium CSEDT7 presented an inhibition percentage close to 20% (Table 2).

### 3.3. Phylogenetic Analysis of Selected Isolates

According to the phylogenetic analysis based on the primers homology to conserved regions of the 16S rRNA gene, the corresponding fragments of the expected nucleotidic dimensions were obtained. When compared with similar fragments on the GenBank database by BLAST, 10 of the 14 isolates of the phyllosphere were found to be related to *Achromobacter insuavis* LMG 26845^T^, CSEDT7 was related to *Luteibacter anthropi*, CCUG 25036^T^, CSFLT6 was related to *Luteibacter yeojuensis* R2A16-10^T^, CRRFLT5 was related to *Rhodococcus cercidiphylli* YIM 65003^T^ and the CRRFLT7 isolate was related to *Pseudomonas parafulva* AJ 2129^T^ (Table 2). The NJ phylogenetic tree was performed using all the isolates with antagonistic activity against the phytopathogenic fungi tested. This analysis showed that the isolates were clustered in four branches, as shown by the significant bootstrap values. The isolates CSFLT4, TSFLT4, TRFLT8, CRRFLT8, CSFLT5, TSFLT2, TSFLT10, CRRFLT6, TSFLT8 and TSFLT3 were clustered with *A. insuavis*, the CRRFLT7 isolate was grouped with the *P. parafulva* group, CSEDT7 and CSFLT6 were clustered with *L. antrophi* and CRRFLT5 was clustered with *Rhodococcus cercidiphylli* (Figure 1). The sequences determined in this study were deposited in the GenBank database.

### 3.4. Inhibitory Effect of the Isolates on the Urediniospores Germination Percentage

The isolates CSEDT7, CSFLT6, CSFLT4, CRRFLT5, CRRFLT7, TRFLT8 and TSFLT2 caused a significant inhibition of urediniospores germination, with percentages between 43 to 86% compared to the uninoculated control, which showed the highest number of germination tubes (Table 3). Within these isolates, TRFLT8 (*A. insuavis*) and CRRFLT7 (*P. parafulva*) showed the best performance, with an inhibition percentage of 86 and 82% (Figure 2), respectively. The behavior observed in these isolates was similar to that shown when the chemical control was applied (91%).

### 3.5. Evaluation of Two Isolates as Coffee Rust Biocontrol under Field Conditions

Regarding the number of leaves, the treatments did not show significant differences between them or compared to the control eight months after the transplant. However, there was a positive trend of CRRFLT7 and the chemical control to maintain a relatively high number of leaves compared to the other treatments (Figure 3C). On the other hand, symptoms of coffee rust were observed using the DI and PDI parameters. While DI shows the percentage of infected leaves per plant, PDI shows the severity percentage of disease damage per plant too. According to the statistical analysis, significant differences (*p*-value < 0.05) in both DI and PDI parameters were observed in plants inoculated with the two isolates and the chemical control compared to the uninoculated plants 8 months after the transplant (Figure 3A,B). The plants inoculated with TRFLT8 (7.7%) and propiconazole (17.9%) showed the same behavior when the disease incidence percentage was evaluated, while the plants inoculated with CRRFLT7 showed a DI of 40.1%. However, the uninoculated control presented an incidence percentage of 77.3%, far outperforming the proven treatments. The severity of the disease resulted in a PDI of 4.4, 15 and 4.6% when propiconazole, CRRFLT7 and TRFLT8 were applied, respectively, compared with the uninoculated control, which showed the greatest severity of damage (29.4%).

## 4. Discussion

This study examines the population of endophytic and epiphytic phyllospheric bacteria in three coffee plant varieties: *typica*, *caturra roja* and *catimor*, which are the most extended varieties in this region of Peru. The *typica* and *caturra* varieties are reported to be popular in many countries [24,25] because of their cup quality attributes. Traditionally, coffee production in Peru is only based on certain varieties such as *typica* and *caturra roja*. Less frequently, *pache*, *mundo novo* and *bourbon* are used, and more recently, *catimor* has been used, which has been widely reported for its tolerance against *H. vastatrix* [26]. The use of the *catimor* variety has not been expanded because its low cup quality is known compared to the susceptible varieties [27]. Despite this, some authors have reported that the quality of *catimor* is not significantly different from that of *typica* or *caturra*, and the quality is not directly related to the variety but also to the climate, altitude, precipitation, harvesting and post-harvesting conditions, among others [28,29,30]. Moreover, some authors have reported that *catimor* varieties showed similarities in organoleptic and sensory features compared to *typica*, *caturra* or *bourbon* [29,31]. The results showed in this work, based on a culture-dependent method, revealed that there were fewer endophytic than epiphytic bacteria; these results were similar to those found by [11], who isolated bacteria from coffee phyllosphere. This could be explained by the fact that only a few bacteria are capable of colonizing internal spaces of leaves that are difficult to access. In terms of variety, symptomatic plants of *typica* showed more epiphytic and endophytic bacteria compared to the other varieties. Moreover, in all the varieties, symptomatic plants have shown more bacterial populations than the asymptomatic ones. Similar results were found by [11] in symptomatic and asymptomatic coffee plants infected with *Xyllela fastidiosa*. However, when bacterial populations were found in asymptomatic and symptomatic plants infected with *M. citricolor*, opposite results were observed. Therefore, foliar bacterial populations may vary depending on the host plant and the phytopathogen that colonizes them [32]. From all the isolates, thirteen epiphytes and only one endophyte, showed antagonistic capabilities against two phytopathogen fungi isolated from coffee. In both cases, it was observed that all the tested bacteria have an inhibition activity against *M. citricolor* better than that against *Colletotrichum* sp. The work [33] reported green ZnO nanobiohybrids against these two phytopathogens. The results showed that the percentage of inhibition depends on the concentration of the nanoparticle more than the fungus itself. Bacteria are capable of producing a wide variety of metabolic compounds with antifungal activity, so the effectiveness is linked to the compound that is produced and not to the fungus it faces. Fourteen potential biocontroller strains from the coffee phyllosphere were identified. Molecular identification using the 16S rRNA gene revealed that the isolates belonged to the *Achromobacter*, *Luteibacter*, *Rhodococcus* and *Pseudomonas* genera. Related studies on olive’s phyllosphere have identified bacteria from the Proteobacteria, Firmicutes and Actinobacteria phyla [34]. Other studies with different crops have found that *Methalobacterium*, *Pseudomonas* and *Sphingomonas* genera were predominant [35]. When all fourteen microorganisms were tested in in vitro inhibition assays against *H. vastatrix*, only three were able to inhibit rust germination effectively, similar to the chemical control. There are many chemicals used for controlling coffee rust, whose efficacy depends mainly on the timing of the application, severity, weather conditions and agronomic crop management, among other factors. Some of the chemical compounds that are used against *H. vastatrix* are dithiocarbamates, mancozeb, copper compounds, strobilurins and triazoles. The in vitro assay used propiconazole as the chemical control, which belongs to the triazole group. This is a systemic fungicide that acts by inhibiting the biosynthesis of ergosterol [36], a key component in the formation of the fungal cell membrane affecting spore germination. The biocontrol exerted by some bacteria against phytopathogenic fungi can be of different types. It can be direct when there is nutritional competition or the production of certain metabolites that inhibit the growth or development of the phytopathogenic fungus. Alternatively, it can be indirect when a substance produced by the bacteria triggers a response in the plant that affects the fungus. *Pseudomonas* spp. were known to produce a wide range of secondary metabolites that indirectly benefit plant growth by inhibiting some phytopathogenic fungi. One mechanism that is widely reported is the ability of *Pseudomonas* spp. to produce siderophores to obtain Fe^+3^ from the rhizosphere, preventing phytopathogens from accessing this nutrient [37]. Moreover, it has been reported that bacterial siderophores affected spore germination and mycelial growth in some phytopathogenic fungi [38]. *Achromobacter xylosoxidans* has been reported as an endophytic bacterium capable of inhibiting the growth and germination of *Fusarium* spp. Its biocontrol ability has been demonstrated in bean plants, where a decrease in the severity of root rot damage was observed [39]. Other species of *Achromobacter* spp. have been reported as plant growth-promoting bacteria and as a biocontrol agent against some phytopathogenic fungi through the production of hydrolytic enzymes or the activation of the plant’s defense by inducing its systemic resistance (SIR). Among these species, *A. insolitus* has shown the capacity to produce cellulases and reduce *Pythium aphanidermatum* growth in in vitro assays [40].

On the field assay, a positive effect of the CRRFLT7 strain and propiconazole on the number of coffee leaves was observed, eight months after the first evaluation. The leaves number is relevant data because *H. vastatrix* attacks coffee leaves, causing defoliation and ultimately leading to the death of the plant. In this context, the effect of the bacteria identified as *Pseudomonas* spp. showed similar behavior to the chemical agent, positioning it as a strain with potential activity to control coffee leaf rust in its early stages. Furthermore, when evaluating the disease index and the percentage of the pathogen disease index (severity), both strains maintained low values, similar to the chemical control. However, there was a better effect observed with the TRFLT8 strain that belongs to the genus *Achromobacter* spp. There was no experimental evidence that confirmed that *P. parafulva* and *A. insuavis* are potentially dangerous pathogens for humans [41,42]. Despite both species being obtained from plants’ rhizospheres, some authors report that *A. insuavis* has also been isolated from chronically infected patients [43]. *Pseudomonas* spp. and *Bacillus* spp. are reported as potential biocontrollers against coffee rust in field trials when applied at the onset of the disease in plants with 5% incidence on leaves [12]. In this work, healthy plants were initially used and became naturally infected, allowing for a better observation of the real effect of bacteria during the phenological development of coffee plants. A decrease in damage was observed in both the applied inoculants and propiconazole compared to the uninoculated control. This indicates that the tested inoculants are potential biocontrollers of coffee leaf rust. They should be used within an integrated crop management approach. Currently, there are various chemical products for controlling coffee leaf rust, which are effective at the onset of the disease. However, there are very few biological products that are marketed to address this problem.

## 5. Conclusions

The use of these bacteria has great economic potential, as the organic coffee market has the best prices in the international market and is subject to fewer fluctuations than the conventional coffee market. The data presented in this study identify two potential biocontrollers not only against leaf rust but also against other phytopathogenic fungi that affect coffee production.

## Figures and Tables

**Figure 1 microorganisms-12-00582-f001:**
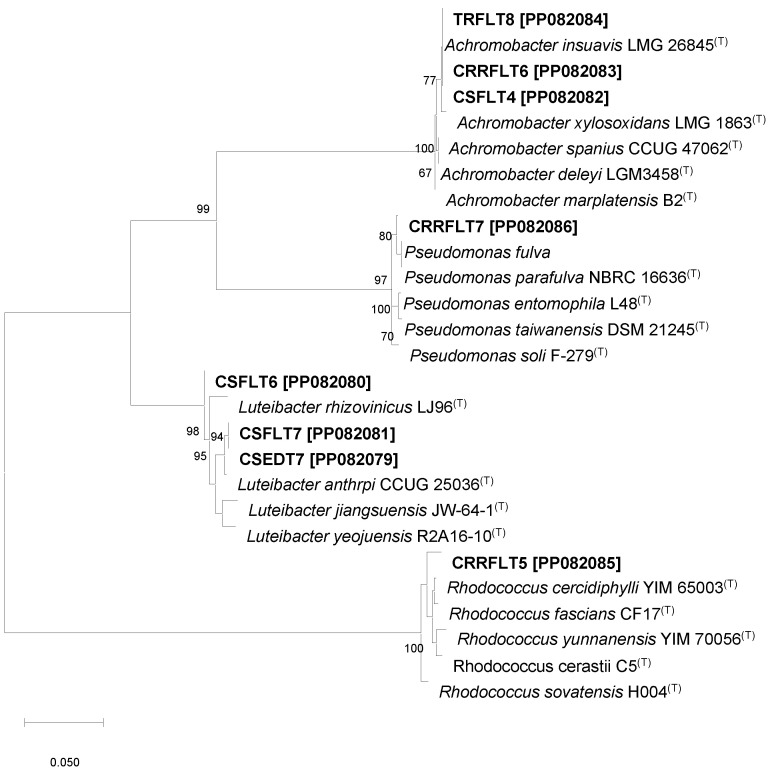
Phylogenetic tree of sequences of the 16S rRNA gene. The phylogenetic reconstruction method of neighbor joining and the distances were calculated in accordance with the Tamura Nei model. The values at the branch points indicate bootstrap support (1000 pseudoreplicates; only values of 50% or above are shown). ^(T)^ The type of that microorganism.

**Figure 2 microorganisms-12-00582-f002:**
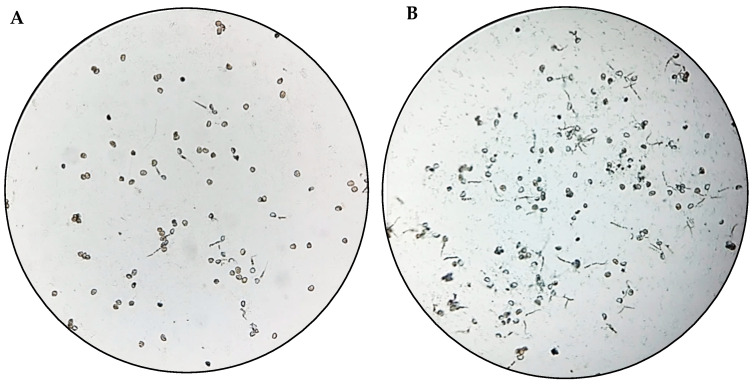
Uredinospores germination of *H. vaxtatrix* (microscopic view at 40× magnification). (**A**) CRRFLT7 treatment, (**B**) negative control.

**Figure 3 microorganisms-12-00582-f003:**
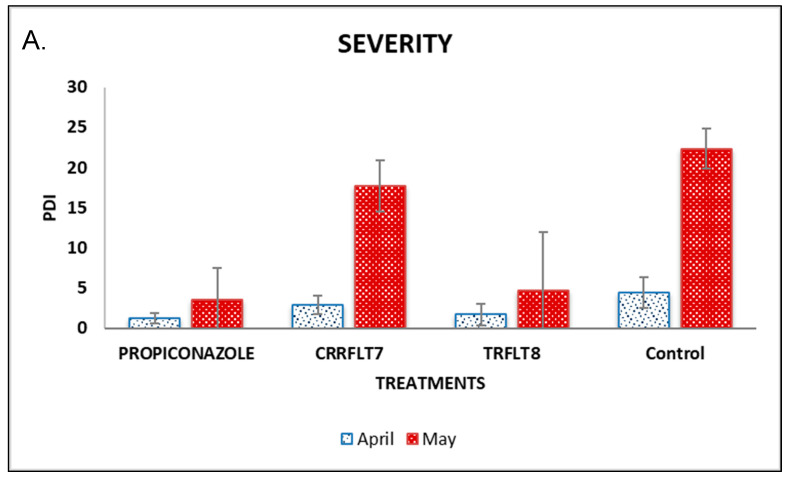
Effect of bacterial inoculation as a biocontroller against coffee rust on field assays. (**A**) Pathogen Disease Index (Severity); (**B**) Disease index (DI%); (**C**) Leaves number per plant for eight months.

**Table 1 microorganisms-12-00582-t001:** Population size of total cultivable epiphytic and endophytic bacteria in coffee leaves of coffee yellow rust symptomatic and asymptomatic plants.

SAMPLE	Endophytes	Epiphytes	
	Aerobic Bacteria (CFU/g Leaf)	Aerobic Bacteria (CFU/g Leaf)	Sporogenic Bacteria (CFU/g Leaf)
CS	13 × 10^3^	13 × 10^4^	23 × 10^2^
CRS	0	40 × 10^3^	50
CRR	0	14 × 10^5^	0
TS	2 × 10	16 × 10^4^	0
TR	21 × 10^5^	52 × 10^9^	28 × 10

CS: *catimor* variety, asymptomatics leaves. CRS: *caturra roja* variety, asymptomatic leaves. CRR: *caturra roja* variety, symptomatic leaves. TS: *typica* variety, asymptomatic leaves. TR: *typica* variety, symptomatic leaves.

**Table 2 microorganisms-12-00582-t002:** Antagonistic activity of bacterial isolates against *M. citricolor* and *Colletrotrichum* spp.

TREATMENTS	Origin	In Vitro Antagonistic Activity Inhibition (%)		Phylogenetic Analysis		
		*M. citricolor*	*Colletrotrichum* sp.	Closely Related Taxa Identified by 16s RNAr GenBank	Similarity (%)	Accession Number
CSEDT7	endophyte	42.2 de	19.8 i	*Luteibacter anthropi* CCUG 25036^(T)^	99.6	FM212561
CSFLT6	epiphyte	35.6 bc	1.1 ab	*Luteibacter anthropi* CCUG 25036^(T)^	98.8	FM212561
CSFLT4	epiphyte	44.4 e	9.8 g	*Achromobacter insuavis* LMG 26845^(T)^	99.6	HF586506
TSFLT2	epiphyte	37.8 bcd	3.8 cd	*Achromobacter insuavis* LMG 26845^(T)^	100	HF586506
TSFLT10	epiphyte	33.3 b	3.0 bc	*Achromobacter insuavis* LMG 26845^(T)^	99.9	HF586506
TSFLT8	epiphyte	33.3 b	3.8 cd	*Achromobacter insuavis* LMG 26845^(T)^	100	HF586506
TSFLT4	epiphyte	33.33 b	13.0 h	*Achromobacter insuavis* LMG 26845^(T)^	99.9	HF586506
CRRFLT6	epiphyte	35.6 bc	3.0 bc	*Achromobacter insuavis* LMG 26845^(T)^	99.7	HF586506
TSFLT3	epiphyte	35.6 bc	5.7 def	*Achromobacter insuavis* LMG 26845^(T)^	99.9	HF586506
CSFLT5	epiphyte	44.4 e	2.4 bc	*Achromobacter insuavis* LMG 26845^(T)^	100	HF586506
CRRFLT8	epiphyte	38.9 cd	7.7 f	*Achromobacter insuavis* LMG 26845^(T)^	99.9	HF586506
CRRFLT5	epiphyte	38.9 cd	12.6 h	*Rhodococcus cercidiphylli* YIM 65003^(T)^	98.7	EU325542
CRRFLT7	epiphyte	38.9 cd	6.2 ef	*Pseudomonas parafulva* AJ 2129^(T)^	99.5	AB060132
TRFLT8	epiphyte	36.7 bc	5.1 de	*Achromobacter marplatensis* B2^(T)^	99.6	EU150134
*control*	--------	0.00 a	0.00 a	------------------------	--------	--------------

Different letters indicate differences between treatments (ANOVA LSD *p* < 0.05). ^(T)^ The type of that microorganism.

**Table 3 microorganisms-12-00582-t003:** Effect of bacterial isolates on the germination of *H. vastatrix* urediniospores.

Treatment	Closed Related Taxa (16s RNAr Gene)	Uredinopores Germination (%)
CSEDT7	*Luteibacter anthropi*	13.87 ab ^†^ (65.32) ^††^
CSFLT6	*Luteibacter rhizovicinus*	14.14 ab (64.64)
CSFLT4	*Achromobacter xylosoxidans*	22.59 b (43.49)
CRRFLT5	*Rhodococcus cercidiphylli*	9.2 a (76.99)
CRRFLT7	*Pseudomonas parafulva*	7.38 a (81.54)
TRFLT8	*Achromobacter insuavis*	5.68 a (85.81)
TSFLT2	*Achromobacter insuavis*	14.28 ab (64.29)
Control químico	-	3.58 a (91.05)
Control-	-	39.98 c

^†^ Different letters indicate differences between treatments (ANOVA LSD *p* < 0.05). ^††^ Numbers in parentheses show the % of inhibition.

## Data Availability

Data available on request due to restrictions.

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
