# Peer review of "Antifungal Activity of Phyllospheric Bacteria Isolated from Coffea arabica against Hemileia vastatrix"

_microorganisms, 2024, doi:10.3390/microorganisms12030582_

Round 1

Reviewer 1 Report

Comments and Suggestions for Authors

This article addresses the antifungal activity of phyllospheric bacteria isolated from Coffea arabica against Hemileia vastatrix, a phytopathogenic fungus causing coffee rust: However, the proposed title does not align with the results shown in this work. The tests conducted against Mycena citricilor and Colletotrichum sp.? lack logical reasoning and were used for the selection of antagonistic bacteria, primarily in the tests on in vitro germination. Moreover, the experiments were limited to the use of two bacteria at a single concentration, restricting the results. The contribution of this work is limited; some experiments exhibit a lack of rigor in methodology, which could affect reproducibility, in addition to several inconsistencies in the presentation. The images and tables require clearer editing for better comprehension.

Specific points to address include:

There is a discrepancy between the number of mycelial growth inhibitory isolates mentioned in the Abstract (15) (line 17) and those presented in Table 1. Describe the abbreviation of the culture mediums (YEM (line 86) and (TGE line 97). It is correct to write Colletotrichum sp., not spp., as it gives the impression that different strains of Colletotrichum were evaluated, whereas the results only show one strain. In this regard, why evaluate the activity against Colletotrichum sp.? Is there a reason for using this particular strain? If so, why wasn't it identified molecularly? Correct the subscripts in the chemical compound formulas (Line 142 to 143). Line 145, why test only at a concentration of 1x108? How can it be asserted that the tests were conducted at the indicated concentration, given the importance of the concentration of antagonistic bacteria in this test? Why not evaluate other concentrations? Clarify the concentration of urediniospores present in the 10 μL, and how many times these experiments were conducted. (Line 149) Mention the procedure used to calculate bacterial abundance (line 176). State the concentration of the fungicide used and why this concentration was chosen (line 151). Improve Table 1 for better understanding and include missing data, such as the results of spore-forming bacteria, or mention them in the results section. Table 3 would benefit from including the full names of the bacteria instead of just their codes.

Comments on the Quality of English Language

English requires minor revision.

Author Response

  1. There is a discrepancy between the number of mycelial growth inhibitory isolates mentioned in the Abstract (15) (line 17) and those presented in Table 1

         line 17 has been changed with 14 strains, instead of 15

  1. Describe the abbreviation of the culture mediums YEM (line 86) and (TGE line 97).

         line 89, 90, 101, 102. The abbreviation of the culture mediums was               described.

  1. It is correct to write Colletotrichum sp., not spp., as it gives the impression that different strains of Colletotrichum were evaluated, whereas the results only show one strain. In this regard, why evaluate the activity against Colletotrichum sp.? Is there a reason for using this particular strain? If so, why wasn't it identified molecularly?

         Colletotrichum spp. was changed to Colletotrichum sp. in the text. It was one strain that was isolated from coffee leaves, and it was identified molecularly - this is informative for a publication that is still a work in progress.

  1. Correct the subscripts in the chemical compound formulas (Line 142 to 143).

      The subscripts in the chemical compound formulas (Line 146 and 149 were corrected)

  1. Line 145, why test only at a concentration of 1x108?

         Line 151: This work aimed to select and characterize phyllospheric bacteria isolated from Coffea arabica with antagonistic features against coffee rust to obtain biocontrollers. In this sense, the dose was not considered as an evaluating factor yet. Bacterial concentration was used according to other studies (Freitas et al., 2023; Fu HZ et al. 2020). Both references were added (line 473 -478)

  1. Clarify the concentration of urediniospores present in the 10 μL

         Concentration was added (line 154)

  1. and how many times these experiments were conducted? (Line 149)

         Done (line 158 y 159)

  1. Mention the procedure used to calculate bacterial abundance (line 176)

          In this work, bacterial abundance was not performed. Instead, quantification of the bacterial population of epiphytes and endophytes was carried out according to the methodologies proposed in the paper. The section Material and Methods now also specify Determination of phyllospheric bacterial populations for improved understanding (line 183)

  1. State the concentration of the fungicide used and why this concentration was chosen (line 151)

         Done (line 157 – 158)

  1. Improve Table 1 for better understanding and include missing data, such as the results of spore-forming bacteria, or mention them in the results section.

         Done. Data of spore-forming bacteria was mentioned in the results section (line 194)

  1. Table 3 would benefit from including the full names of the bacteria instead of just their codes.

         Done

Reviewer 2 Report

Comments and Suggestions for Authors

Article "Antifungal Activity of Phyllospheric Bacteria Isolated From 2 Coffea arabica Against Hemileia vastatrix", By studying the endophytic and epiphytic phyllospheric varieties of coffee leaf rust sensitive varieties Typica and Caturra roja and the rust-resistant varieties Catimor, and screening bacteria resistant to yellow coffee rust. The endophytic and epiphytic phyllospheric strains were identified by 16s RNA. The results showed that CRRFLT7 and TRFLT8 had good inhibition rate on the germination of urediniospores, and field experiments also showed that these two strains showed good control effect on yellow coffee rust, which provided materials for the biological control of yellow coffee rust. The approach and comparative analysis of the present study is sound but still it needs some  revisions. The outcomes are interesting and the study will contribute to the knowledge in the respective area. However, the language revision should be painstakingly done. Some specific comments are as followed:

1. The inhibition rates of CRRFLT7 and TRFLT8 on the germination of summer spores described in the abstract were 81% and 82%, respectively, but the inhibition rates of TRFLT8 and CRRFLT7 on the germination of summer spores described in part 3.4 were 86% and 82%, respectively, please check.

2. In the "introduction" part, please unify the description method of hectares. There are several different descriptions of "hectares", "h" and "ha" in the article.

3. 2.5 Part 143 lines "0.6g、0.3g" and other numbers and units are missing Spaces. Similar points to modify the full text or the number and unit combination format for the full text unified.

Comments on the Quality of English Language

English language is fine.

Author Response

  1. The inhibition rates of CRRFLT7 and TRFLT8 on the germination of summer spores described in the abstract were 81% and 82%, respectively, but the inhibition rates of TRFLT8 and CRRFLT7 on the germination of summer spores described in part 3.4 were 86% and 82%, respectively, please check.

        The error was corrected.

  1. In the "introduction" part, please unify the description method of hectares. There are several different descriptions of "hectares", "h" and "ha" in the article.

         In the introduction all texts that appear hectares, ha, or h have all been corrected to ha

  1. 5 Part 143 lines "0.6g、0.3g" and other numbers and units are missing Spaces. Similar points to modify the full text or the number and unit combination format for the full text unified.

         Done

Reviewer 3 Report

Comments and Suggestions for Authors

Review of the article: „Antifungal activity of phyllospheric bacteria isolated from Coffea arabica against Hemileia vastatrix

Manuscript ID - microorganisms-2784712

In my opinion the idea of the research was interesting an very important. The experiments were well planned and performed. The manuscript is prepared carefully and obtained results are promising. In my opinion the manuscript can be accepted for publication in Microorganism. However, I have some critical remarks that should be considered by authors of the articles. Below I have presented detailed comments about the manuscript.

Detailed comments

Abstract  - abstract is generally well prepared. I do not have critical comments.

Introduction

The information presented in introduction justify the idea of the research, which in my opinion was very interesting and important.

Methodology

Lines 74 and 75 – the authors should shortly explain how the coffee varieties are classified as tolerant or resistant to coffee yellow rust;

 Lines 81-100 – the composition of YEM and TGE media should be presented. Was it really necessary to incubate the plates for 15 days (I expect that some colonies appeared on the agar plates after shorter time of incubation). Why spore forming bacteria were grown for only 24 hours?

Line 101 – why these strains (M. citricolor and Colletotrichum spp. – whole names should be presented) were used for investigation of antifungal activity of bacterial isolates. The coffee yellow rust is caused by Hemileia vastatrix.

Line 104 – the composition of PDA agara should be presented.

The unit 2.4 – the expected size of PCR products should be mentioned. I understand that different strains were investigated and some differences in the length of PCR product could be observed, but the approximate size is known and should be presented.

Line 160 - CRRFLT7 and 160 TRFLT8 isolates were selected for the field experiment. But are the authors sure that they are safe for people (consumers and farmers).

Results

Table 2 – the inhibition % - the result should be presented with an accuracy of no more than one decimal place. In my opinion at least short information about these bacteria species should be presented in the manuscript. First of all it should be checked if they are safe for people and wild animals.

Section 3.4 – would it be possible to present a picture with the result of this experiment

Discussion and conclusions – generally well prepared.

However,  the authors should consider presenting conclusions as a separate “chapter”

Lines 292 – 294 – “[32] reported green ZnO nanobiohybrids against 292 these two phytopathogens. Results showed that the percentage of inhibition depends on 293 the concentration of the nanoparticle more than the fungus itself.” – I am not sure if this part of discussion fits to the rest of manuscript.

Final opinion – minor revision

Author Response

Methodology

Lines 74 and 75 – the authors should shortly explain how the coffee varieties are classified as tolerant or resistant to coffee yellow rust;

Done (line 75 and 76)

 Lines 81-100 – the composition of YEM and TGE media should be presented. Was it really necessary to incubate the plates for 15 days (I expect that some colonies appeared on the agar plates after shorter time of incubation). Why spore forming bacteria were grown for only 24 hours?

Composition of YEM and TGE  mediums were included. Some microorganisms like actinomycetes, for example would appear after 15 days. Spore-forming bacteria were grown only for 24 h because references specify that period for evaluation.

Line 101 – why these strains (M. citricolor and Colletotrichum spp. – whole names should be presented) were used for investigation of antifungal activity of bacterial isolates. The coffee yellow rust is caused by Hemileia vastatrix.

  1. citricolor and Colletotrichum sp. were phytopathogenic fungi that affect coffee plants. Previous inhibition assays before H. vastatrix were performed like a screening experiment because these fungi were easier to work than H. vastatrix (which is an obligate biotrophic fungi). H. vastatrix material (urediniospores) for the inhibition assays was difficult to collect and maintain, so that was our strategy to - select the potential bacterial strains for the inhibition assays using urediniospores of yellow rust

Line 104 – the composition of PDA agar should be presented.

Composition of PDA medium was included

The unit 2.4 – the expected size of PCR products should be mentioned. I understand that different strains were investigated and some differences in the length of PCR product could be observed, but the approximate size is known and should be presented.

Done, line 123

Line 160 - CRRFLT7 and 160 TRFLT8 isolates were selected for the field experiment. But are the authors sure that they are safe for people (consumers and farmers).

We checked those strains, and there were no reports of pathogens. Only one of them was reported as an opportunistic pathogen. We checked references and many authors used these species as plant growth-promoting rhizobacters in field experiments.

Results

Table 2 – the inhibition % - the result should be presented with an accuracy of no more than one decimal place. In my opinion at least short information about these bacteria species should be presented in the manuscript. First of all it should be checked if they are safe for people and wild animals.

Table 2. decimal numbers were rounded to one decimal place. Information about the strains were added (line 421-424)

Section 3.4 – would it be possible to present a picture of the result of this experiment

Done

Discussion and conclusions – generally well prepared.

However,  the authors should consider presenting conclusions as a separate “chapter”

Done

Lines 292 – 294 – “[32] reported green ZnO nanobiohybrids against 292 these two phytopathogens. Results showed that the percentage of inhibition depends on 293 the concentration of the nanoparticle more than the fungus itself.” – I am not sure if this part of the discussion fits the rest of the manuscript.

We think there is a relation to the work because bacteria could have metabolites that inhibit phytopathogen growth. Additionally, in that study,  researchers worked against strains of Colletotrichum sp.and Mycena citricolor – which is the same as what we have done with this research.

Round 2

Reviewer 1 Report

Comments and Suggestions for Authors

The authors made most of the requested corrections.

Comments on the Quality of English Language

Good English